# ⛁PROMPTHUB: ENHANCING MULTI-PROMPT VISUAL IN-CONTEXT LEARNING WITH LOCALITY-AWARE FUSION, CONCENTRATION AND ALIGNMENT

**Tianci Luo**[1,*]**, Jinpeng Wang**[2,*,†]**, Shiyu Qin**[1]**, Niu Lian**[2]**,**
**Yan Feng**[3]**, Bin Chen**[2,†]**, Chun Yuan**[1,†]**, Shu-Tao Xia**[1]
[1]Tsinghua Shenzhen International Graduate School, Tsinghua University
[2]Harbin Institute of Technology, Shenzhen
[3]Meituan, Beijing
`ltc25@mails.tsinghua.edu.cn; wangjp26@gmail.com;`
`chenbin2021@hit.edu.cn; yuanc@sz.tsinghua.edu.cn`

## ABSTRACT

Visual In-Context Learning (VICL) aims to complete vision tasks by imitating pixel demonstrations. Recent work (Wang et al., 2025) pioneered prompt fusion that combines the advantages of various demonstrations, which shows a promising way to extend VICL. Unfortunately, the patch-wise fusion framework and model-agnostic supervision hinder the exploitation of informative cues, thereby limiting performance gains. To overcome this deficiency, we introduce PromptHub, a framework that holistically strengthens multi-prompting through locality-aware fusion, concentration and alignment. PromptHub exploits spatial priors to capture richer contextual information, employs complementary concentration, alignment, and prediction objectives to mutually guide training, and incorporates data augmentation to further reinforce supervision. Extensive experiments on three fundamental vision tasks demonstrate the superiority of PromptHub. Moreover, we validate its universality, transferability, and robustness across out-of-distribution settings, and various retrieval scenarios. This work establishes a reliable locality-aware paradigm for prompt fusion, moving beyond prior patch-wise approaches. Code is available at `https://github.com/luotc-why/ICLR26-PromptHub`.

## 1 INTRODUCTION

Foundation models like GPT (Brown et al., 2020), Llama (Touvron et al., 2023), Gemini (Team et al., 2023) and Flamingo (Alayrac et al., 2022) have demonstrated the emerging ability of demonstration-based prompt learning, aka In-Context Learning (ICL) (Dong et al., 2024; Zheng et al., 2023; Yang et al., 2023), which further facilitates their versatility in various tasks. The basic idea of ICL (Hendel et al., 2023; Wei et al., 2023; 2022; Jiang et al., 2024) is to prompt models with some demonstrative input-output pairs in addition to the query input, which can enhance the answer robustness. The reliablity of ICL has been thoroughly validated (Shin et al., 2022; Yoo et al., 2022; Dai et al., 2023; Von Oswald et al., 2023). Recently, Visual ICL (VICL) (Bar et al., 2022; Wang et al., 2023b) have also become a popular topic, where pixel-space in-painting is the native paradigm.

Choosing appropriate prompts is critical in VICL. Many recent works (Zhang et al., 2023; Sun et al., 2025; Xu et al., 2024; Zhu et al., 2025) focused on optimizing retrievers to select better suited prompts, and Zhang et al. (2024) incorporated visual prompt tuning to enhance the robustness of VICL. Nature Language Processing (NLP) literature (Shi et al., 2022; Gao et al., 2024) suggests multiple prompts can enhance ICL with mitigated bias and richer context, offering insights essential for advancing VICL. Yet visual backbones like MAE-VQGAN typically restrict inputs to a single prompt, rendering multi-prompting non-trivial. In practice, there are two heuristic strategies for extending single-prompting to multi-prompting, namely downscaling (Wang et al., 2023a) and ensemble (Sun et al.,

---

*Equal contribution.
†Corresponding authors.

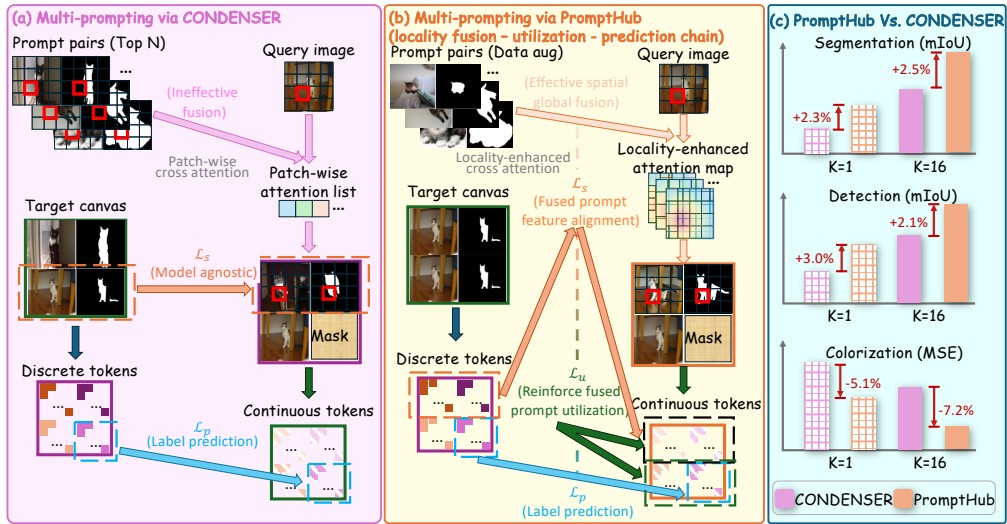

Figure 1: (a) CONDENSER performs patch-wise fusion to fuse composite prompt, while leveraging model-agnostic supervision signals at the input level. (b) PromptHub transcends CONDENSER by enforcing a locality-aware chain that unifies fusion-utilization-prediction. It aligns spatial priors into coherent prompt representations, reinforces the backbone's concentration on fused cues, and integrates label prediction to maintain the integrity of VICL pipeline. (c) Comparison of CONDENSER and PromptHub across three tasks under both single-prompting and multi-prompting configurations.

2025). Building upon this, CONDENSER (Wang et al., 2025) was the first to adopt prompt fusion (Liu et al., 2023b), integrating useful information from multiple prompts into a fused prompt, as illustrated in Figure 1(a). However, its patch-wise fusion strategy results in substantial underuse of valuable cues, while model-agnostic supervision remains insufficient. Moreover, discrepancies between fused prompt and query pair may compels backbone to distrust fused representation, falling back on its own capacity for inference. This is precisely the situation we aim to avoid.

We break these limits by proposing PromptHub, which aims at **(i)** integrate precise knowledge from diverse prompts, **(ii)** mitigate fused prompt's discrepancies to encourage the backbone's effective trust and reliance on it, and **(iii)** ultimately yield superior VICL predictions. To achieve this, we propose a locality-aware fusion framework together with three cooperative learning objectives, as illustrated in Figure 1(b). Specifically, we introduce a locality prior that applies spatially decaying weights radiating from the current patch, thereby enhancing accurate feature extraction. This design allows the fusion process to retain a global receptive field while alleviating the adverse effects of border noise. During the optimization of PromptHub, we design three complementary objectives: **(i)** an end-to-end semantic integrity loss to promote high-quality prompt fusion by aligning fused exemplars with query semantics, as semantically closer prompt generally benefit VICL; **(ii)** a utilization loss that mitigates discrepancies between the fused prompt and the query pair, thereby promoting the backbone's trust and reliance on the fused representation for imitation learning; and **(iii)** a label prediction loss, retained from CONDENSER, which serves as the base supervision to preserve VICL's contextual prediction behavior. Additionally, we preliminarily explored VICL-oriented data augmentation strategies to enhance the robustness of PromptHub. These designs realize chain-wide enhancements for VICL.

We evaluate PromptHub on segmentation, detection, and colorization. As shown in Figure 1(c), extensive experiments demonstrated its superiority to state-of-the-art baselines. We also demonstrate PromptHub's promising resource efficiency, transferability and robustness to various prompt retrieval strategies including random selection. Comprehensive ablations on diverse learning objectives, data augmentation techniques, and locality fusion, confirming the success of our paradigm design. We further visualize the fused prompts, which validates the reliablity of PromptHub. These findings strongly support the efficacy of PromptHub, highlighting the significance of our approach in VICL.

To sum up, we make the following contributions.

- We introduce a locality-enhanced fusion strategy that balances spatial locality and receptive field, enabling more comprehensive extraction of effective information.

- We propose three complementary learning objectives that collaboratively enhance prompt fusion quality, strengthen prompt concentration, and improve contextual prediction, further reinforced with VICL-specific data augmentation.
- Extensive experiments show PromptHub's efficacy beyond state-of-the-art techniques. Promising results also suggest that it is transferable across domains and robust to prompt retrieval, establishing a reliable competitive new solution in VICL.

## 2 RELATED WORKS

### 2.1 LARGE VISION MODELS

The field of computer vision has witnessed substantial advancements, driven by abundant foundational models (Chang et al., 2022; Wang et al., 2023c; Oorloff et al., 2025; Qiu et al., 2025; Yu et al., 2026). LVM (Bai et al., 2024), an auto-regressive generative model, effectively converted visual information into language-like visual sentences and improved understanding capabilities. MAE and Point-MAE (He et al., 2022; Pang et al., 2022), utilizing a reconstruction strategy, established unified visual architectures across various downstream tasks in 2D and 3D domains. The ability for ICL has also been demonstrated within foundation models, as researchers employed specialized training methodologies (Bar et al., 2022; Fang et al., 2024; Wang et al., 2023b; Wei et al., 2025; Yue et al., 2024; Yu et al., 2025; Yue et al., 2023) to endow these models with superior in-context learning capabilities, thus providing a robust foundation for the domain of Visual ICL (VICL).

### 2.2 VISUAL IN-CONTEXT LEARNING VIA IN-PAINTING

MAE-VQGAN (Bar et al., 2022) and Painter (Wang et al., 2023a) serves as crucial in-painting backbones for VICL, with copious works building upon and enhancing this framework. Existing work (Zhang et al., 2023; Sun et al., 2025; Xu et al., 2024) primarily focuses on retrieval to obtain better prompt. Sun et al. (2025) studied prompt spatial arrangement, testing eight configurations and reporting improved results through voting. Zhang et al. (2024) pioneered visual prompt tuning (Pfeiffer et al., 2020; Hu et al., 2021; Liu et al., 2023a; Bahng et al., 2022), adding a noise border to prompts. PANICL (Zhang et al., 2025) employs a training-free k-nearest-neighbor fusion integrates multiple prompts to alleviate the bias inherent in single prompt. PICO (Jiang et al., 2025) reformulates personalized vision problem under the VICL paradigm and exhibits clear advantages. Hojel et al. (2024) focused on identifying task vector activates backbone to optimize VICL process. CONDENSER (Wang et al., 2025) leveraged prompt composition Li et al. (2024) to aggregate informative cues from multiple prompts. However, the patch-wise information aggregation strategy in CONDENSER (Wang et al., 2025) exhibits inherent limitations, and its supervision over exemplar quality remains insufficiently comprehensive. Motivated by these, we propose a locality fusion scheme coupled with three cooperative objectives, establishing a more reliable paradigm for prompt fusion.

## 3 METHOD: PROMPTHUB

### 3.1 PROBLEM FORMULATION AND METHOD OVERVIEW

Given a prompt database, $\mathcal{D} = \{P_i\}_{i=1}^{|\mathcal{D}|}$, where each prompt comprises an image-label pair. The pixel-level retriever $\mathcal{R}$ identifies top-$N$ similar prompt pairs $\mathcal{P} = \{P_n = (X_n, Y_n)\}_{n=1}^N$, for a given query image $X_q \in \mathbb{R}^{H \times W \times 3}$. Following previous settings, we adopt MAE-VQGAN, configured with patch size of 16 and feature dimension of $D$, as the backbone. Under general setting, the closest pair $P_1 = (X_1, Y_1) \in \mathbb{R}^{H \times 2W \times 3}$ provides the prompt. We concatenate prompt $P_1$ with the query image $X_q$ to construct the canvas $S_1 = \begin{bmatrix} X_1 & Y_1 \\ X_q & [M] \end{bmatrix}$, where $[M]$ denotes the mask need to be recovered. The backbone output at the masked location corresponds to the $X_q$'s predicted label $\hat{Y}_q$.

In our framework, $N$ prompt pairs $\mathcal{P}$ and query image $X_q$ are processed by the query-adaptive PromptHub module, producing fused features $F_{X_f}, F_{Y_f} \in \mathbb{R}^{\frac{H}{16} \times \frac{W}{16} \times D}$. $F_{X_f}, F_{Y_f}$ and query image

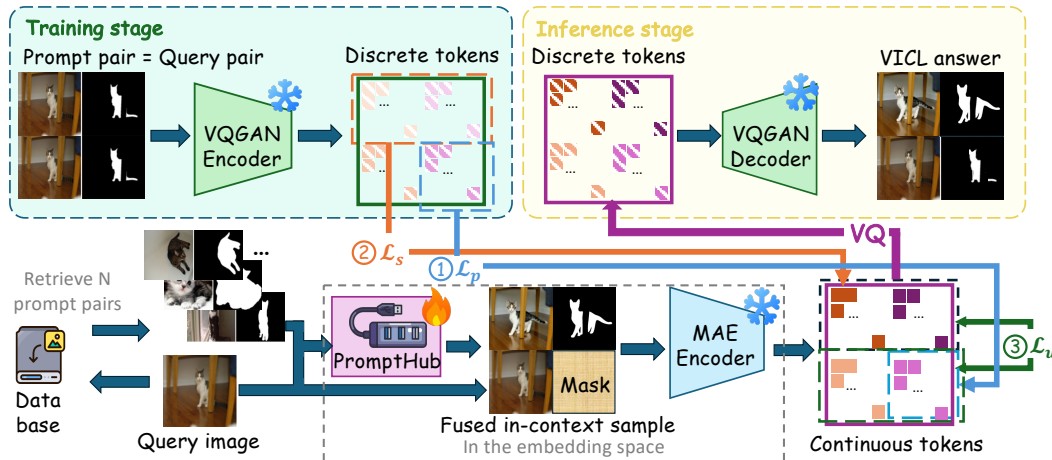

Figure 2: The training and inference framework of PromptHub based on MAE-VQGAN.

features $F_{X_q}$, mask features $F_{[M]}$ are concatenated into the canvas $S_f = \begin{bmatrix} F_{X_f} & F_{Y_f} \\ F_{X_q} & F_{[M]} \end{bmatrix}$. Then $S_f$ is passed through MAE-VQGAN, excluding the patch embedding, to generate the VICL answer.

## 3.2 PROMPTHUB MODULE DESIGN

To expand the receptive field during fusion while mitigating the impact of boundary noise, we employ a locality-enhanced prompt fusion strategy. PromptHub locally fuse $\mathcal{P}$ into a unified prompt pair $(F_{X_f}, F_{Y_f})$ in the embedding space. The workflow is shown in Figure 3.

We first process images $(X_q, \mathcal{P})$ by embedding layer to yield $E_{X_q}, E_{X_{1:N}}, E_{Y_{1:N}} \in \mathbb{R}^{\frac{H}{16} \times \frac{W}{16} \times D}$.

Subsequently, we deploy a self-attention transformation $\mathrm{SA}(\cdot)$ to align the query and prompts to similar patterns, thereby generating the resultant features $F_{X_q}, F_{X_{1:N}}, F_{Y_{1:N}}$.

Thereafter, we use a query-adaptive locality-enhanced cross-attention to extract spatial information from prompt pairs features $(F_{X_{1:N}}, F_{Y_{1:N}})$, which achieves the fused exemplar features $(F_{X_f}, F_{Y_f})$.

We define the locality prior as a probability distribution controlled by the hyper-parameter $\sigma$. To effectively represent this locality distribution, we employ either Gaussian prior or Laplacian prior, with the choice governed by a hyperparameter:

$$\psi(h, w, x, y) = \begin{cases} \exp\left(-\frac{(x-h)^2 + (y-w)^2}{2\sigma^2}\right), & \text{Locality prior = Gaussian prior} \\ \exp\left(-\frac{\sqrt{(x-h)^2 + (y-w)^2}}{\sigma}\right), & \text{Locality prior = Laplacian prior} \end{cases}. \quad (1)$$

For each query image token $F_{X_q}[h, w]$, it has a specific locality matrix $\Psi_{h,w}$ centered at $(h, w)$:

$$\Psi_{h,w} = \begin{bmatrix} \psi(h, w, 1, 1) & \cdots & \psi(h, w, 1, \frac{W}{16}) \\ \vdots & \ddots & \vdots \\ \psi(h, w, \frac{H}{16}, 1) & \cdots & \psi(h, w, \frac{H}{16}, \frac{W}{16}) \end{bmatrix}. \quad (2)$$

During the VICL inference phase, no matched query label $Y_q$ is available for constructing fused prompt label $F_{Y_f}$. However, the specific correspondence still exists between prompt images $X_{1:N}$ and prompt labels $Y_{1:N}$. Therefore, we share the prompt images features $F_{X_{1:N}}$ as the key in the attention mechanism, compute the generalized attention scores, and subsequently perform localized weighting to procure locality-enhanced attention weights $A_{h,w} \in \mathbb{R}^{N \times \frac{H}{16} \times \frac{W}{16}}$ for $F_{X_f}[h, w]$:

$$A_{h,w} = \mathrm{softmax}\left(\frac{\left(F_{X_q}[h, w] \times W_Q\right) \times \left(F_{X_{1:N}} \times W_K\right)^\top}{\sqrt{D}} \cdot \Psi_{h,w}\right), \quad (3)$$

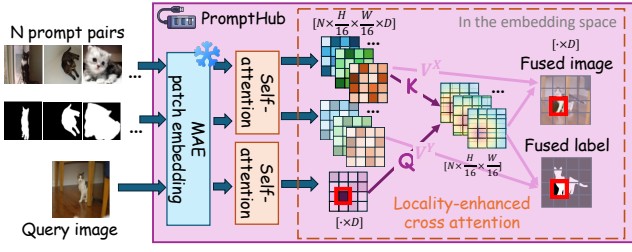

Figure 3: **PromptHub module design.** $N$ prompt pairs and query image are embedded into the MAE patch space, where locality-enhanced fusion integrates spatially cues into a fused prompt aligned with query's informative content.

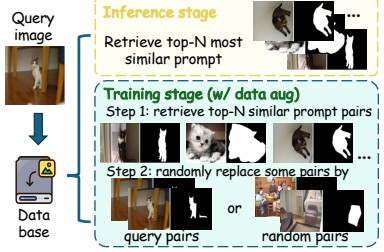

Figure 4: **Data augment of PromptHub.** In training, the top-$N$ pairs are randomly substituted with either query pairs or random pairs.

which $\cdot$ denotes element-wise multiplication, $\times$ denotes matrix multiplication, and $W_Q, W_K \in \mathbb{R}^{D \times D}$ represent the projection layers for mapping $Q$ and $K$ in the attention mechanism.

Ultimately, we multiply the locality-enhanced attention weights with the features $F_{X_{1:N}}$ and $F_{Y_{1:N}}$ through linearly transformed to obtain the fused prompt pair features $F_{X_f}, F_{Y_f} \in \mathbb{R}^{\frac{H}{16} \times \frac{W}{16} \times D}$. $W_{V^X}, W_{V^Y} \in \mathbb{R}^{D \times D}$ denote linear layers in attention mechanism for image and label, respectively.

$$F_{X_f}[h, w] = A_{h,w} \times (F_{X_{1:N}} \times W_{V^x}), \quad F_{Y_f}[h, w] = A_{h,w} \times (F_{Y_{1:N}} \times W_{V^Y}). \quad (4)$$

### 3.3 LEARNING OBJECTIVES

We introduce three complementary learning objectives to guide fusion module's training, collectively strengthening *"fusion–utilization–prediction"* closed-loop for robust VICL, as illustrated in Figure 2.

**(i) Ensuring label prediction performance.** Following CONDENSER and InMeMo, we also adopt a label prediction loss as the fundamental objective to preserve VICL's contextual prediction behavior. Without this base supervision, parameterized VICL paradigms cannot function properly. Upon deriving the fused in-context sample $S_f$, we propagate it through the MAE encoder, generating a canvas of continuous tokens $\begin{bmatrix} T^c_{X_f} & T^c_{Y_f} \\ T^c_{X_q} & T^c_{[M]} \end{bmatrix} \in \mathbb{R}^{\frac{2H}{16} \times \frac{2W}{16} \times D}$. These tokens are calibrated during pretraining to correspond with the VQGAN codebook space. Simultaneously, We construct the target canvas $S_q = \begin{bmatrix} X_q & Y_q \\ X_q & Y_q \end{bmatrix} \in \mathbb{R}^{2H \times 2W \times 3}$ by integrating the query pair as a prompt pair and process it through the VQGAN encoder, obtaining the corresponding discrete tokens $\begin{bmatrix} T^{d(1)}_{X_q} & T^{d(1)}_{Y_q} \\ T^{d(2)}_{X_q} & T^{d(2)}_{Y_q} \end{bmatrix} \in \{1, 2, ..., N_c\}^{\frac{2H}{16} \times \frac{2W}{16}}$ from the codebook. Here, $N_c$ denotes the size of the codebook space, with $N_c = D$. $T^{d(1)}_{X_q}$ and $T^{d(1)}_{Y_q}$ represent the discrete tokens output as prompt, while $T^{d(2)}_{X_q}$ and $T^{d(2)}_{Y_q}$ correspond to the discrete tokens output as query. To optimize the label prediction results, we align the bottom-right portion $T^c_{[M]}$, which will be reconstructed by the VQGAN decoder, with the target $T^{d(2)}_{Y_q}$ using a cross-entropy loss. Here, $\mathcal{L}_p$ denotes the loss function for *label prediction*:

$$\mathcal{L}_p = -\mathbb{E}_{(h,w) \sim \mathcal{U}([1, \frac{H}{16}] \times [1, \frac{W}{16}])} \log T^c_{[M]} \left[ h, w, T^{d(2)}_{Y_q} \right]. \quad (5)$$

**(ii) Fused-prompt feature alignment.** The backbone tends to produce accurate predictions when exposed to the same prompt as the query. We employ a cross-entropy alignment between the continuous tokens derived from the fused prompt pair $(T^c_{X_f}, T^c_{Y_f})$ and the discrete tokens corresponding to the query pair as a prompt $(T^{d(1)}_{X_q}, T^{d(1)}_{Y_q})$ to make fused prompt pair closely approximate the query pair. The semantic integrity loss for improved *fusion* is denoted as $\mathcal{L}_s$.

$$\mathcal{L}_s = -\mathbb{E}_{(h,w) \sim \mathcal{U}([1, \frac{H}{16}] \times [1, \frac{W}{16}])} \left( \log T^c_{X_f}[h, w, T^{d(1)}_{X_q}] + \log T^c_{Y_f}[h, w, T^{d(1)}_{Y_q}] \right). \quad (6)$$

**(iii) Enhance fused prompt utilization.**     Owing to discrepancy between fused prompt and query pair, backbone may regard useful prompt as unreliable and instead rely on its own capacity. We employ a cosine-similarity loss $\mathcal{L}_u$, designed to reduce the dissimilarity between query pair $(T_{X_q}^c, T_{[M]}^c)$ and fused prompt $(T_{X_f}^c, T_{Y_f}^c)$, thereby enhancing the backbone's *utilization* of fused prompt.

$$\mathcal{L}_u = -\mathbb{E}_{(h,w)\sim\mathcal{U}([1,\frac{H}{16}]\times[1,\frac{W}{16}])} \left( \cos\left( T_{X_f}^c\left[h,w\right], T_{X_q}^c\left[h,w\right] \right) + \cos\left( T_{Y_f}^c[h,w], T_{[M]}^c[h,w] \right) \right). \tag{7}$$

We adopt $\lambda$ and $\gamma$ to balance diffent losses. Let $\theta$ denote the parameters of PromptHub. The ultimate synergistic optimization objective is formulated as:

$$\min_{\theta}\ \mathcal{L}_p + \lambda\mathcal{L}_s + \gamma\mathcal{L}_u. \tag{8}$$

## 3.4 RETRIEVE SCHEME FOR DATA AUGMENT

In the inference phase, we consistently retrieve the top-$N$ most similar prompt pairs $\mathcal{P} = \{P_n\}_{n=1}^N$ from the database $\mathcal{D}$, utilizing the most relevant raw prompt pairs for improved VICL.

During training, we employ a data augmentation strategy to enhance two regularization objectives' effect. Based on the retrieved top-$N$ prompt pairs $\mathcal{P} = \{P_n\}_{n=1}^N$, we might replace some prompt pairs $P_n$ with either query pairs $P_q = (X_q, Y_q)$ or randomly retrieved pairs $P_r$, as shown in Figure 4.

**(i) Substitute with query pair to better utilize the fused prompt.**     Under typical settings, defining prompt pair $P_1$ as query pair $P_q$ generally yields minimal discrepancy. To this end, we replace current prompt pairs $P_n$ with query pairs $P_q$ with probability $p_q$. This substitution establishes a purified learning objective that minimizes discrepancy as much as possible, hence enhancing $\mathcal{L}_u$.

**(ii) Substitute with random pair to enhance PromptHub's robustness.**     With probability $p_r$, we substitute prompt pair $P_n$ with a randomly retrieved pair $P_r$, introducing a controlled level of noise that enhances $\mathcal{L}_s$ and PromptHub's stability. This technique guarantees when high-quality prompts are unavailable during inference, PromptHub retains its capacity to achieve robust VICL results.

# 4 EXPERIMENTS

## 4.1 EXPERIMENTAL SETUP

**Downstream Tasks and Datasets.**     To ensure a fair comparison, we employ three well-established tasks foreground segmentation, single-object detection, and colorization along with their associated datasets, within the domain of VICL. For **foreground segmentation**, we employ Pascal-$5^i$ (Shaban et al., 2017), which consists of four folds, with each fold containing data from five different classes. We conduct experiments across all folds and analyze the results by presenting the mean intersection over union (mIoU) for each fold. In the case of **single-object detection**, we utilize the Pascal VOC2012 (Everingham et al., 2015) dataset, also employing mIoU as the evaluation metric. For the **coloring task**, we randomly select 50,000 images from the ImageNet-1K ILSVRC2012 (Russakovsky et al., 2015) training set, with 50 images chosen from each of the 1,000 classes to form the label portion of our training set. The 50,000 images from the validation set of ImageNet-1K ILSVRC2012 are used as the label portion of our test set. We convert training set and test set label portion to grayscale images, which served as the input queries. We use MSE as the evaluation metric.

**Implementation Details.**     We adopt MAE-VQGAN (Bar et al., 2022) as the backbone architecture and utilize Prompt-SelF's (Sun et al., 2025) pixel-level retriever for prompt retrieval. During training, we use the training set as the database for prompt pairs while also employing the training set as the query. In the testing phase, the validation set serves as the query collection, while the training set acts as the database. The input image resolution to the model is $224 \times 224$, with each sub-image having a resolution of $112 \times 112$. We utilized Gaussian prior as the default locality prior.

**Training Configurations.**     We employed SGD optimizer with a learning rate initialized at 0.04, which decays according to cosine annealing warm restarts scheduler. For segmentation and detection tasks, training is performed for 100 epochs, while coloring task requires 10 epochs. The corresponding $\sigma$ values for foreground segmentation, object detection, and colorization tasks are 0.65, 0.5, and 2.5, respectively. Hyper-parameter $\lambda$ is set to 0.5, and $\gamma$ is set to 0.2. The experiments were performed on single 80G A100 GPUs with a batch size of 16.

Table 1: PromptHub performance is compared with different baselines in three downstream tasks foreground segmentation (**Seg**.), single-object detection (**Det**.), and image colorization (**Col**.). The results for $N = 1, 16$, representing the cases with 1 and 16 prompts respectively, are listed separately. The highest results are denoted in **bold**, while the suboptimal results are indicated in *italics*.

| Model | Seg. (mIoU ↑) | | | | | Det. (mIoU ↑) | Col. (MSE ↓) |
|---|---|---|---|---|---|---|---|
| | Fold-0 | Fold-1 | Fold-2 | Fold-3 | Mean | | |
| *Zero-Shot* | | | | | | | |
| Random (Bar et al., 2022) | 28.66 | 30.21 | 27.81 | 23.55 | 27.56 | 25.45 | 0.67 |
| UnsupPR (Zhang et al., 2023) | 34.75 | 35.92 | 32.41 | 31.16 | 33.56 | 26.84 | 0.63 |
| Prompt-SelF (Sun et al., 2025) | 35.69 | 38.25 | 35.86 | 33.37 | 35.79 | 28.08 | 0.63 |
| *Retriever Training* | | | | | | | |
| SupPR (Zhang et al., 2023) | 37.08 | 38.43 | 34.40 | 32.32 | 35.56 | 28.22 | 0.63 |
| Partial2Global (Xu et al., 2024) | 38.81 | 41.54 | 37.25 | 36.01 | 38.40 | 30.66 | 0.58 |
| *PEFT* | | | | | | | |
| InMeMo (Zhang et al., 2024) | 41.65 | 47.68 | 42.43 | 40.80 | 43.14 | 43.21 | - |
| *Task Vectors* | | | | | | | |
| VTV (Hojel et al., 2024) | 38.00 | 37.00 | 33.00 | 32.00 | 33.50 | - | - |
| *Prompt Fusion* | | | | | | | |
| CONDENSER$_{N=1}$ (Wang et al., 2025) | 42.13 | 50.31 | 42.20 | 41.90 | 44.14 | 43.22 | 0.560 |
| CONDENSER$_{N=16}$ (Wang et al., 2025) | *45.53* | *52.06* | *44.33* | *44.58* | *46.63* | *44.64* | 0.539 |
| PromptHub$_{N=1}$ *(Ours)* | 43.26 | 50.75 | 43.83 | 42.82 | 45.17 | 44.51 | *0.533* |
| PromptHub$_{N=16}$ *(Ours)* | **45.93** | **53.12** | **45.44** | **46.74** | **47.81** | **45.59** | **0.503** |

## 4.2 COMPARISON WITH STATE-OF-THE-ARTS

**Baselines.** We compare our method against comprehensive state-of-the-art approaches built on the MAE-VQGAN framework. Our competitors are categorized into four groups: (1) Zero-shot methods, including MAE-VQGAN (Bar et al., 2022) and UnsupPR (Zhang et al., 2023) and Prompt-SelF (Sun et al., 2025), which do not require additional retriever training; (2) Methods that necessitate retriever training, such as SupPR (Zhang et al., 2023) and Partial2Global (Xu et al., 2024); (3) Approach that leverages prompt tuning, exemplified by InMeMo (Zhang et al., 2024); (4) Method of finding and utilizing the task vector VTV (Hojel et al., 2024). (5) Method that employs prompt fusion, CONDENSER (Wang et al., 2025), to enable multi-prompt VICL, with comparisons reported under both single-prompting and multi-prompting settings.

**(i) Performance on Standard Tasks.** Table 1 demonstrates that PromptHub achieves consistent improvements across all tasks under both single-prompt and multi-prompt settings. In single-prompt scenario, PromptHub surpasses CONDENSER by 2.3%, 3.0%, and 5.1% on segmentation, detection, and colorization, respectively. Under multi-prompt scenario, it further attains gains of 2.5%, 2.1%, and 7.2% on same tasks. PromptHub's output visualization is discussed further in the appendix.

**(ii) Performance on Domain Adaption Task.** In real-world applications, the data for inference may undergo domain adaptation compared to the training data. Thus, testing the transferability of different VICL schemes is crucial. We trained the PromptHub on the COCO-$5^i$ (Lin et al., 2014) using the same settings as previous works (Wang et al., 2025; Sun et al., 2025; Zhang et al., 2024), and evaluate it on the Pascal-$5^i$. As shown in Table 2, PromptHub

Table 2: Transferability evaluation. We train models on COCO-$5^i$ and test on Pascal-$5^i$.

| Model | Seg. (mIoU ↑) | | | | |
|---|---|---|---|---|---|
| | Fold-0 | Fold-1 | Fold-2 | Fold-3 | Mean |
| Prompt-SelF | 40.13 | 42.14 | 37.84 | **38.52** | 39.66 |
| InMeMo | 38.74 | 43.82 | 40.45 | 37.12 | 40.03 |
| CONDENSER$_{N=1}$ | *40.39* | 44.54 | 40.23 | 36.33 | 40.37 |
| CONDENSER$_{N=16}$ | 40.37 | *44.85* | *41.03* | 35.84 | 40.52 |
| PromptHub$_{N=1}$ | 40.36 | 45.24 | 40.43 | *37.94* | *41.00* |
| PromptHub$_{N=16}$ | **42.69** | **46.71** | **41.97** | 37.31 | **42.17** |

demonstrates substantially larger improvements than other baselines, outperforming CON-DENSER by 4.1% in the multi-prompt setting, highlighting the strong transferability.

**(iii) Performance under the multi-prompting scenario.** To validate the scalability of PromptHub, we compare it with CONDENSER under various $N$, specifically 1, 2, 4, 8, 16, and 32. In addition, we report results under down-sampling$_{N=2,N=7}$ (Zhang et al., 2023) and answer-level$_{N=16}$ (Sun et al., 2025). The

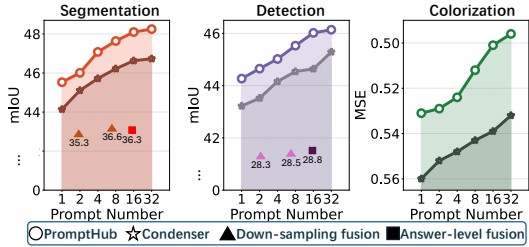

Figure 5: Performance comparison with baselines in multi-prompt VICL scenario.

Table 3: Ablation study of PromptHub. The best are marked in **bold** and second-best in *italic*.

| # | Model | Seg. (mIoU ↑) | | | | | Det. (mIoU ↑) |
|---|---|---|---|---|---|---|---|
| | | Fold-0 | Fold-1 | Fold-2 | Fold-3 | Mean | |
| (0) | PromptHub$_{N=1}$ | 43.26 | 50.75 | 43.83 | 42.82 | 45.17 | 44.51 |
| (1) | PromptHub$_{N=16}$ | *45.93* | **53.12** | 45.44 | **46.74** | **47.81** | **45.59** |
| *Effectiveness of Learning Objectives* | | | | | | | |
| (2) | w/o $\mathcal{L}_u$ $_{N=1}$ | 42.71 | 51.14 | 42.78 | 42.41 | 44.76 | 43.45 |
| (3) | w/o $\mathcal{L}_u$ $_{N=16}$ | 45.54 | 52.25 | 44.59 | 44.47 | 46.71 | 44.83 |
| (4) | w/o $\mathcal{L}_s$ $_{N=1}$ | 42.23 | 50.52 | 42.29 | 42.16 | 44.30 | 43.12 |
| (5) | w/o $\mathcal{L}_s$ $_{N=16}$ | 44.72 | 51.77 | 43.57 | 43.30 | 45.84 | 44.27 |
| (6) | w/o $\mathcal{L}_p$ $_{N=1}$ | 8.51 | 10.13 | 9.46 | 8.33 | 9.11 | 13.23 |
| (7) | w/o $\mathcal{L}_p$ $_{N=16}$ | 9.41 | 13.44 | 12.29 | 10.62 | 11.44 | 12.87 |
| *Effectiveness of Locality-Enhanced Fusion* | | | | | | | |
| (8) | w/ Laplacian Prior$_{N=1}$ | 43.74 | 50.93 | 43.51 | 43.05 | 45.31 | 43.93 |
| (9) | w/ Laplacian Prior$_{N=16}$ | **46.42** | *52.87* | **45.45** | *46.16* | *47.72* | *45.47* |
| (10) | Global Fusion$_{N=1}$ | 41.77 | 49.04 | 42.69 | 40.73 | 43.55 | 41.86 |
| (11) | Global Fusion$_{N=16}$ | 41.91 | 50.45 | 43.76 | 42.43 | 44.64 | 42.49 |
| (12) | Convolution-Based Fusion$_{N=1}$ | 42.56 | 50.15 | 42.79 | 42.52 | 44.51 | 43.83 |
| (13) | Convolution-Based Fusion$_{N=16}$ | 45.28 | 51.68 | 45.34 | 45.51 | 46.95 | 45.07 |
| *Effectiveness of Data Augment Technique* | | | | | | | |
| (14) | w/o Data Augment$_{N=1}$ | 43.11 | 51.22 | 43.17 | 42.34 | 44.96 | 43.52 |
| (15) | w/o Data Augment$_{N=16}$ | 45.84 | 52.01 | 44.83 | 45.60 | 47.07 | 45.06 |

Figure 6: The visualization of the fused prompt pair after passing through the VQGAN decoder.

experimental results demonstrate our approach not only improves performance as $N$ increases, but also consistently surpasses other baselines by a large margin, as shown in Figure 5.

## 4.3 MODEL ANALYSIS

For a comprehensive ablation study, we designed several variants, as summarized in Table 3, where Variants (0) – (1) correspond to the canonical configurations.

**(i) Effectiveness of Learning Objectives.** To comprehensively evaluate the contributions of each learning objective, we conducted an ablation analysis by individually removing the three objectives. The experimental results demonstrate *"fusion-utilization-prediction"* objectives are mutually complementary, and omitting any of them leads to performance degradation in multi-prompt VICL. In particular, the primary objective, label prediction $\mathcal{L}_p$, is indispensable for preserving VICL's contextual prediction behavior; *without it, the training-based VICL paradigm with additional parameters cannot function effectively*. Meanwhile, $\mathcal{L}_s$ and $\mathcal{L}_u$ act as crucial regularization terms, ensuring fused exemplars' quality and the backbone's effective utilization. *The absence of either damages the pipeline in VICL and results in mediocre performance.*

**(ii) Effectiveness of Locality-Enhanced Prompt Fusion.** We compare locality-enhanced fusion with global fusion, patch-wise fusion (CONDENSER (Wang et al., 2025)), and convolution-based

fusion, where the latter replaces the spatial prior with convolutional transformations. *Notably, locality-enhanced fusion can be viewed as a higher-level framework, within which global fusion and patch-wise fusion emerge as two complementary instantiations, corresponding to larger and smaller values of the locality parameter σ, respectively.* As shown in Table 3, both types of locality priors achieve superior performance. *Upon observation, maintaining an appropriate balance between global receptive fields and spatial locality proves essential. The locality accords with the fusion principle that enriches information capture while mitigating long-range noise.*

**(iii) Effectiveness of Data Augment Technique.** We conducted experiments under scenarios without data augmentation, only utilizing the top-$N$ prompt pairs for fusion during training, as illustrated in Variants (14) – (15). The results indicate that removing data augmentation diminishes the performance of PromptHub in VICL tasks, confirming the effectiveness of data augmentation. It better reinforce fused prompt utilization and enhances noise resistance.

**(iv) Impact on Different Retrievers.** *Exploring better prompt retrieval and investigating multi-prompt fusion are two orthogonal research directions, while the fusion plugin can be adapted to different retrievers.* We investigated performance of PromptHub using different retrievers, as presented in Figure 7. We evaluated four types of retrievers: random selection (Bar et al., 2022), UnsupPR (Zhang et al., 2023), SupPR (Zhang et al., 2023) and Pixel-Level retriever (Sun et al., 2025). Experimental results demonstrate PromptHub is more effective than CON-DENSER across all retrieval schemes, further highlighting its generalizability. Additionally, the performance of our method is influenced by the choice of retriever; pixel-level retrievers consistently deliver best results, underscoring the alignment between pixel-level retrieval and locality-aware design philosophy.

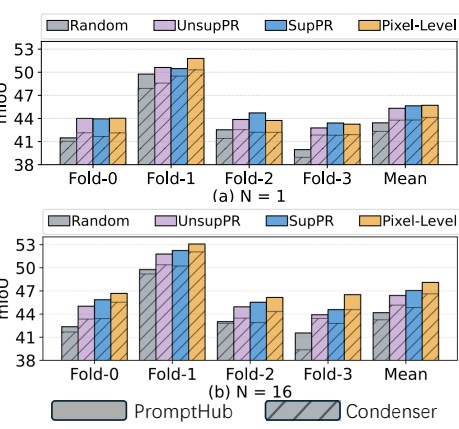

Figure 7: Comparison of PromptHub and CONDENSER across different retrieval.

**(v) Transferability on Unseen Tasks.** We evaluate cross-task transferability by training all models solely on segmentation (Pascal-5, four folds) and directly testing them on detection (Pascal VOC 2012) without any fine-tuning. We compare PromptHub with the CONDENSER baseline using their released checkpoints, and report results in Table 4.

PromptHub consistently surpasses Condenser in this challenging unseen-task setting. With $N = 16$, PromptHub achieves a +1.83% mIoU gain, indicating that our locality-aware fusion captures more robust and transferable visual cues than the patch-wise fusion used in Condenser. *We note that although the overall performance is strong, the training process is not fully task-agnostic. Since the model is trained to re-construct segmentation masks, a domain gap naturally emerges when transferring to bounding box detection, which leads to a certain degree of performance drop.*

Table 4: Transferability experiment (unseen task) where both CONDENSER and PromptHub are trained on segmentation and evaluated on detection.

| Method | Det. (mIoU ↑) | | | | |
|---|---|---|---|---|---|
| | **Fold-0** | **Fold-1** | **Fold-2** | **Fold-3** | **Mean** |
| Condenser$_{N=1}$ | 38.15 | 35.70 | 35.49 | 30.31 | 34.91 |
| Condenser$_{N=16}$ | 41.25 | 36.66 | 37.86 | 39.02 | 38.70 |
| PromptHub$_{N=1}$ | 41.59 | 36.25 | 37.71 | 32.15 | 36.93 |
| PromptHub$_{N=16}$ | 43.40 | 38.55 | 39.52 | 40.66 | 40.53 |

**(vi) Performance Evaluation under Spatial Misalignment.** Spatial misalignment between prompts and queries may negatively affect prompt fusion performance. To evaluate performance under position shifts, we conducted an experiment where query pairs were horizontally flipped and retrained to simulate severe spatial misalignment between the retrieved prompts and the query image. We compared the performance drop of CON-DENSER and PromptHub under the perturbed conditions in Table 5. PromptHub is substantially more robust to spatial misalignment than CONDENSER. Its locality-aware fusion mitigates the sensitivity to positional shifts that affects CONDENSER's patch-wise fusion. In addition,

Table 5: Comparison of standard and perturbed mIoU under spatial misalignment, along with the corresponding performance drops.

| Method | Standard mIoU | Perturbed mIoU | Performance Drop |
|---|---|---|---|
| Condenser$_{N=1}$ | 44.14 | 42.36 | -1.78 |
| Condenser$_{N=16}$ | 46.63 | 45.24 | -1.39 |
| PromptHub$_{N=1}$ | 45.17 | 44.01 | -1.16 |
| PromptHub$_{N=16}$ | 47.81 | 47.15 | -0.66 |

increasing number of prompts $N$ further reduces misalignment effects by improving chance of encountering better-aligned prompt pairs.

## 4.4 DISCUSSION: WHAT DOES PROMPTHUB LEARN?

In Figure 6, we present visualizations of Prompt-SelF, as well as fusion samples reconstructed through the VQGAN decoder for CONDENSER$_{N=16}$ and PromptHub$_{N=16}$. Given that this visualization relies on reconstructed outputs, some bias may be inevitably introduced. We observe that in Prompt-SelF, label prediction often tends to be highly similar to the retrieved prompt label, leading to poor performance when the retrieved label show little similarity to ground-truth answer.

The fusion results of CONDENSER appear as noisy black-and-white patterns, which may be attributed to its model-agnostic feature matching and patch-wise attention that fail to generate smooth representations, offering only heuristic contributions to performance. *In contrast, the fused prompts produced by PromptHub exhibit significantly better visual quality, with fused prompts showing high similarity to the query pairs and smooth textures, thereby confirming the advantages of the locality-aware design and offering a more reliable and trustworthy solution for prompt fusion in VICL.* Furthermore, we quantitatively compare the mIoU between fused prompt labels and query labels for CONDENSER and PromptHub, as shown in Table 6. PromptHub$_{N=16}$ achieves a 72% higher similarity to the ground truth compared with CONDENSER$_{N=16}$, demonstrating PromptHub produces higher-quality and more semantically coherent fused prompts. Although fused prompts may exhibit a gap from realistic images due to lack of fidelity constraints in decoding stage, our primary goal is to guide VICL inference rather than to generate photorealistic images.

Table 6: Comparison the mIoU between the fused prompt labels and the query labels across methods to evaluate semantic alignment.

| Method | Seg. (mIoU ↑) | | | | |
|--------|--------|--------|--------|--------|------|
| | Fold-0 | Fold-1 | Fold-2 | Fold-3 | Mean |
| Condenser$_{N=1}$ | 18.93 | 29.73 | 24.26 | 27.94 | 25.22 |
| Condenser$_{N=16}$ | 14.27 | 24.56 | 18.85 | 20.56 | 19.56 |
| PromptHub$_{N=1}$ | 21.25 | 37.61 | 35.01 | 29.79 | 30.92 |
| PromptHub$_{N=16}$ | 25.22 | 43.22 | 36.07 | 30.41 | 33.73 |

## 5 CONCLUSIONS

In this work, we introduced PromptHub, a interpretability paradigm realizes the chain-wide enhancements *"locality fusion–utilization–prediction"* for multi-prompt VICL. PromptHub balances spatial locality with global receptive fields, supervises the quality of fused samples, and enhances the backbone's utilization on integrated prompts. Extensive experiments across diverse tasks demonstrate clear improvements over previous methods. Furthermore, PromptHub's superior transferability, robustness and generalizability further highlight its potential for extensive implementation in diverse scenarios. We finally visualize the fused prompts, the results outperform patch-wise scheme and provide stronger interpretability for prompt fusion methods.

**Acknowledgments.** We sincerely thank the anonymous reviewers and chairs for their efforts and constructive suggestions, which have greatly helped us improve the manuscript. This work is supported in part by the National Key R&D Program of China (2022YFB4701400/4701402), the National Natural Science Foundation of China under grants 624B2088, 62576122, 62301189, 62571298, and in part by the SSTIC Grants KJZD20230923115106012, KJZD20230923114916032, and GJHZ20240218113604008.

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

## A    SCOPE OF LLM USAGE

To remain compliant with responsible LLM usage protocols, we limited the scope of LLMs to improving readability and grammar. Every scientific contribution, including the conceptual development, experimental design, and analytical validation, was independently carried out and confirmed by the authors, and we retain complete responsibility.

## B    FUTURE WORKS AND LIMITATIONS

### B.1    WHITE-BOX DEPENDENCY

Like CONDENSER, PromptHub requires access to the backbone's parameters and gradients to train the fusion module, even though the backbone itself is frozen. This design has been instrumental in ensuring the framework's success and robustness in its current applications. But this makes scaling to very large models or closed-source models challenging, as full gradients may be inaccessible or too costly. While this enables superior performance, extending prompt fusion to black-box or gradient-free settings is a key direction for future work.

### B.2    EXTENDING APPLICABILITY TO LINGUISTIC AND MULTI-MODAL DOMAINS

PromptHub is designed for VICL tasks with constrained inputs, utilizing positional correspondences between query and label image patches for locality-enhanced prompt fusion. Building on its success in the visual domain, future work will expand its scope to multi-modal scenarios by exploring generalized mechanisms that effectively align visual and linguistic modalities, enabling broader applicability and integration.

## C    PRELIMINARY: MAE-VQGAN

As described in Figure 8, MAE-VQGAN (Bar et al., 2022), comprising the MAE (He et al., 2022) and VQGAN components, serves as a backbone for VICL through an in-painting approach. Given an example and query for the current task, MAE-VQGAN is treated as a versatile model capable of solving several image-to-image tasks.

During the pre-training phase, the model is trained on a dataset CVF, where each image is constructed from multiple sub-images, proceeding the masked reconstruction task. This process fine-tune the MAE encoder to align the distances with its the VQGAN's codebook space. In the inference phase, a in-context sample is fed into the MAE encoder, and the corresponding content from the VQGAN's codebook space is obtained, which is then passed to the VQGAN decoder for generating the output.

We utilizes the pre-trained parameters of MAE-VQGAN, freezing its parameters throughout the entire process.

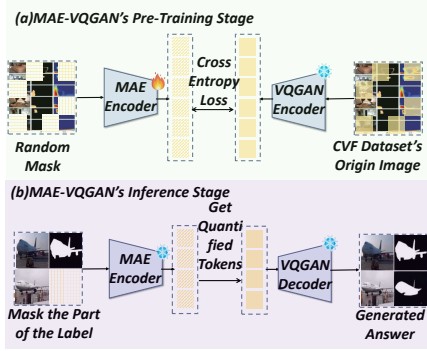

Figure 8:    Introduction to MAE-VQGAN (Bar et al., 2022): (a) In the pre-training stage, MAE (He et al., 2022) is trained to enhance its inference capability through a masked reconstruction task on CVF dataset. (b) In the inference stage, the prompt pair is placed above, with the query positioned below, and both are fed into the model for generative processing.

## D  INFERENCE TIME AND GPU OVERHEAD

As shown in Table 7, we compare the inference time and GPU usage of PromptHub with other baselines. The time for retrieving prompt pair is not included in the inference time. All methods that require only prompt pair retrieval are categorized under the MAE-VQGAN class. It can be observed that the time overhead of our approach increases only modestly compared to other methods, with GPU usage growing at approximately 30MB per prompt pair. Therefore, PromptHub is resource-efficient. This further confirms the lightweight nature of the plug-in PromptHub based on prompt fusion, which incurs only minimal additional computational and GPU overhead. The study underscores the practical feasibility of deploying this approach in real-world scenarios, offering an effective and resource-efficient solution.

Table 7: Comparison of the inference time and GPU overhead between PromptHub and baselines.

| Method | Inference Time (ms/query) | GPU Cost (MB/query) |
|---|---|---|
| MAE-VQGAN | 51.26 | 416.14 |
| InMeMo | 54.28 | 497.50 |
| Prompt-SelF$_{N=16}$ | 984.62 | 441.75 |
| CONDENSER$_{N=1}$ | 59.17 | 565.42 |
| CONDENSER$_{N=16}$ | 66.61 | 1021.86 |
| PromptHub$_{N=1}$ | 63.14 | 569.88 |
| PromptHub$_{N=16}$ | 70.40 | 1032.50 |

## E  ANALYSIS OF HYPERPARAMETER

### E.1  ANALYSIS OF HYPERPARAMETER $\sigma$

The hyperparameter $\sigma$ influences the neighborhood range selected by PromptHub. When $\sigma \to 0$, the selected neighborhood consists solely of the content of the current $(h, w)$ token. As $\sigma \to \infty$, the selected neighborhood encompasses global information, equivalent to the standard cross-attention. As shown in Figure 9, extremely large or small values of $\sigma$ result in either insufficient emphasis on local information or neglect of global information. Moreover, the optimal $\sigma$ value varies across tasks. For high-level and low-level tasks, $\sigma = 0.5$ and $\sigma = 2.5$ are both reasonable choices, respectively.

## F  EXPERIMENTAL ANALYSIS OF QUERY-CONDITIONAL SIGMA

We design a straightforward query-conditioned sigma mechanism to investigate the impact of adaptive $\sigma$ for the same task. Specifically, we average the embedding dimension of the query [batchsize, patch-number, embeddim], apply a linear layer, and use a sigmoid activation to constrain the sigma value within $(0,1)$. We report its performance on segmentation and detection tasks.

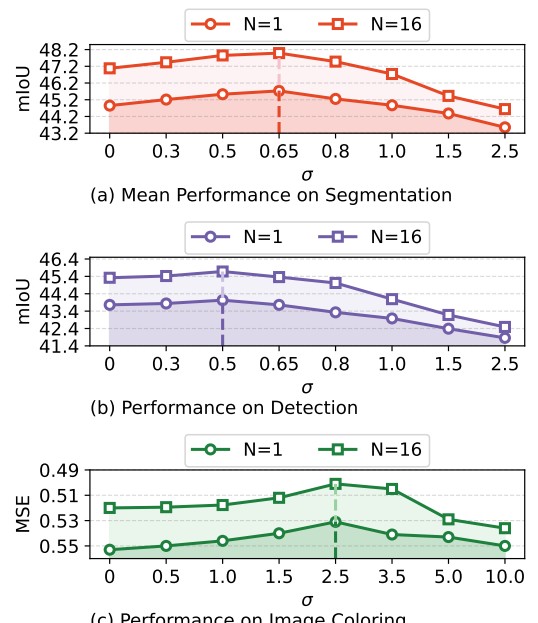

Figure 9: Evaluation of PromptHub's performance on three tasks across varying values of $\sigma$.

Table 8: Comparison of results between query-conditioned sigma and hyperparameter sigma.

| Method | Fold-0 | Fold-1 | Fold-2 | Fold-3 | Mean | Det |
|---|---|---|---|---|---|---|
| PromptHub$_{N=1}$(query-adaptive sigma) | 43.79 | 51.93 | 44.56 | 43.18 | 45.86 | 44.25 |
| PromptHub$_{N=16}$(query-adaptive sigma) | 46.44 | 52.97 | 45.66 | 46.89 | 47.99 | 45.41 |
| PromptHub$_{N=1}$(hyperparameter sigma) | 43.26 | 50.75 | 43.83 | 42.82 | 45.17 | 44.51 |
| PromptHub$_{N=16}$(hyperparameter sigma) | 45.93 | 53.12 | 45.44 | 46.74 | 47.81 | 45.59 |

As shown in Table 8, employing a simple learnable $\sigma$ within the same task yields limited improvements. This suggests that more sophisticated spatially varying priors are required, which we leave for future exploration.

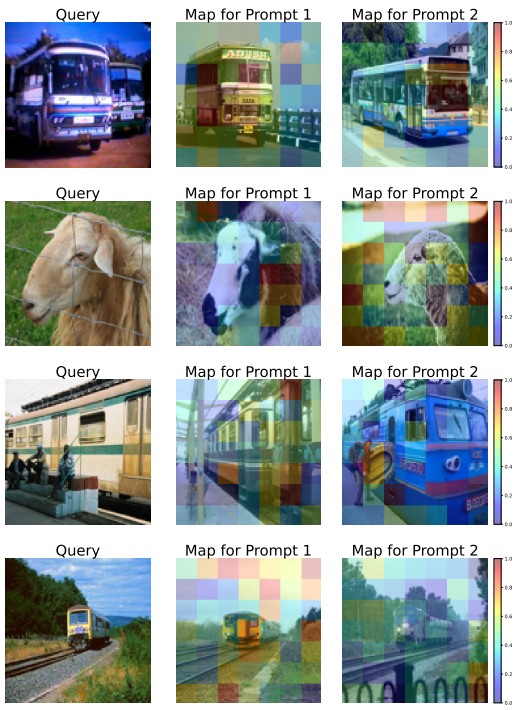

Figure 10: Visualization of attention map for $N = 2$.

## G  EXPLORING MULTI-OBJECTIVE SEGMENTATION

Table 9: Results of Multi-Objective Segmentation Experiments.

| Method | Fold-0 | Fold-1 | Fold-2 | Fold-3 | Mean |
|---|---|---|---|---|---|
| SupPR | 26.85 | 32.73 | 33.48 | 28.40 | 30.37 |
| InMeMo | 28.13 | 38.31 | 37.94 | 33.08 | 34.37 |
| PromptHub$_{N=16}$ | 38.56 | 46.54 | 45.34 | 39.23 | 42.41 |

We further report the numerical results on multi-object segmentation, using a subset filtered by annotations. As shown in Table 9, on the complex task of multi-objective segmentation, our PromptHub model achieves an average mIoU that surpasses the strongest competitor, InMeMo, by approximately 23.4%. This demonstrates that our approach maintains strong transferability in challenging tasks and exhibits robust generalization capability.

## H  MORE VISUALIZATION

### H.1  VISUALIZATION OF VICL ANSWER VIA PROMPTHUB

As illustrated in Figure 11, PromptHub consistently outperforms prior baselines across all three tasks. In particular, the segmentation and colorization results demonstrate that the predictions generated by PromptHub exhibit smoother textures, which further substantiates the advantages of the locality-aware paradigm. Moreover, the ability of PromptHub to strengthen multi-prompt VICL highlights its potential to drive more comprehensive progress in this domain.

### H.2  VISUALIZATION OF ATTENTION MAP

As shown in Figure 10, we visualize the attention map for prompt fusion with $N = 2$, demonstrating that PromptHub effectively focuses on regions corresponding to the query image. The attention score for the current patch is computed as the normalized result of its attention score from all query patches.

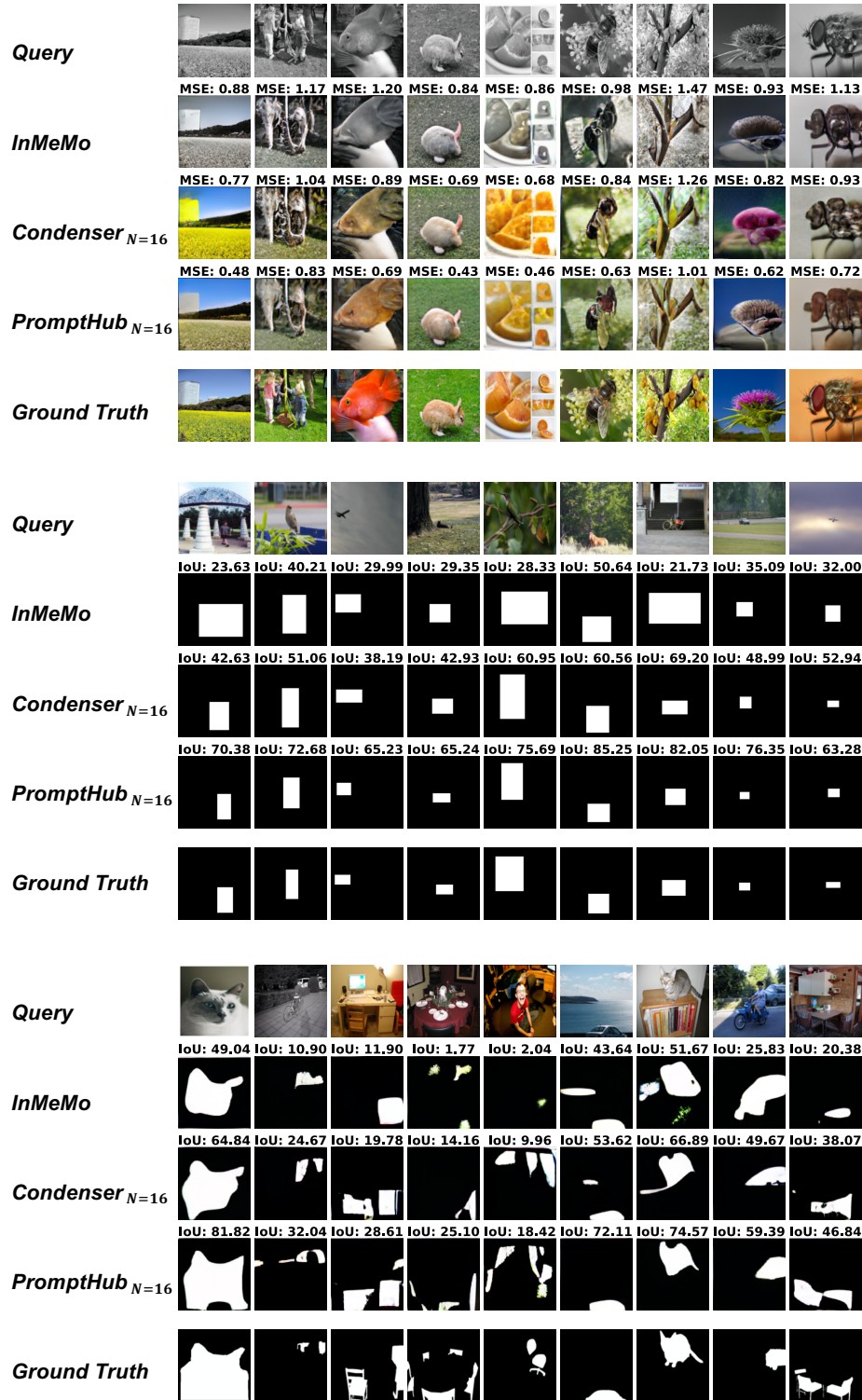

Figure 11: Comparative visualization of our method against the existing state-of-the-art method for Foreground Segmentation and Single-Object Detection and Colorization tasks.

