# OpenReview forum: "PromptHub: Enhancing Multi-Prompt Visual In-Context Learning with Locality-Aware Fusion, Concentration and Alignment"
_ICLR.cc/2026/Conference — ICLR 2026 Poster_

### Official Review · Reviewer_3dRG · 2025-10-31

**Soundness:** 3
**Presentation:** 3
**Contribution:** 3
**Rating:** 6
**Confidence:** 3

**Summary:**

In this paper, the authors propose PromptHub, a locality-aware multi-prompt fusion framework for Visual In-Context Learning (VICL). By integrating three cooperative losses (semantic alignment, utilization, and prediction), PromptHub addresses the shortcomings of patch-wise fusion methods like CONDENSER.

**Strengths:**

1. The overall idea is well-motivated, as the paper clearly identifies the limitation of patch-wise fusion in existing multi-prompt VICL methods and introduces a locality-aware fusion mechanism that intuitively enhances spatial coherence and prompt utilization.

2. Extensive experiments showing consistent gains across segmentation, detection, and colorization.

**Weaknesses:**

1. The claim that PromptHub ‘establishes an interpretable paradigm’ is somewhat overstated. Moreover, the analysis of interpretability remains superficial, lacking quantitative evidence or deeper reasoning about what the fused representations capture.

2. The visualization results are not entirely convincing.  For instance, in some examples (e.g., prompt 1), background regions such as the sky or ground areas from unrelated prompts (e.g., prompt 2) are also highlighted, suggesting that the fusion may still mix irrelevant spatial cues rather than providing truly interpretable alignment.

3. The replacement probabilities $p_q$，$p_r$ are not justified, and the approach lacks ablations on different augmentation ratios. It’s unclear whether improvements come from the locality-aware fusion or the added stochastic augmentation.

4. The paper does not explicitly discuss the limitations of the proposed method.

**Questions:**

Could the authors provide some failure cases or qualitative examples where PromptHub does not perform well, and analyze the possible reasons behind these failures?

---

> ### Author Response · Authors · 2025-11-25
> **Author Response (1/2)**
>
> Dear reviewer `3dRG`,
>
> We thank you for your review and valuable suggestions! We appreciate your positive assessment that **the overall idea is well motivated**. Below, we address each of your suggestions in detail.
>
> > Q1: Interpretability analysis should be supported with quantitative evaluation.
>
> **A1:**
> To rigorously measure the alignment between the fused prompt and the query, we computed the IoU between the fused prompt label and the query's ground-truth label. This metric objectively evaluates how well the fused knowledge corresponds to the target object.
>
> As shown in the table below, PromptHub significantly outperforms Condenser:
> | Method             | Fold-0 | Fold-1 | Fold-2 | Fold-3 | Mean    |
> |--------------------|--------|--------|--------|--------|--------|
> | Condenser$_{N=1}$  | 18.93  | 29.73  | 24.26  | 27.94  | 25.22 |
> | Condenser$_{N=16}$ | 14.27  | 24.56  | 18.85  | 20.56  | 19.56  |
> | PromptHub$_{N=1}$  | 21.25  | 37.61  | 35.01  | 29.79  | 30.92 |
> | PromptHub$_{N=16}$ | 25.22  | 43.22  | 36.07  | 30.41  | 33.73  |
>
> PromptHub(N=16) achieves an average IoU of **33.73**, representing an approximate **72%** improvement over Condenser(N=16) (19.56).
>
> These results quantitatively confirm that PromptHub produces fused prompts more semantically aligned with the query targets.
>
> *Interpretable*: For clarity, we have revised the terminology in the manuscript by replacing *interpretable* with *reliable* to better reflect the intended meaning.
>
> We have added a discussion of this issue in Section 4 of the revised manuscript.
>
> > Q2: The fused prompt still contains elements not relevant to the primary query object ('bus').
>
> **A2:** We appreciate the reviewer's detailed observation regarding the attention maps. **However, the highlighting of background regions (e.g., sky or ground) is intentional and desirable behavior, rather than "bugs".**
>
> Our rationale is as follows:
>
> 1. PromptHub aims to produce a fused prompt that captures the **"full content"** of the query image, irrespective of foreground or background distinctions. While the downstream task may be object-centric (e.g., segmenting a bus), the VICL process benefits from a holistic scene analogy. If the query image contains "sky" or "road" in the background, the fusion module should attend to similar background semantic regions in the prompts (e.g., prompt 1's sky matching the query's sky).
>
> 2. *Focusing solely on the central object may hinder many VICL tasks. Tasks such as image colorization and image denoising do not revolve around salient objects, and thus relying on object-centric cues would restrict the applicability of the VICL backbone.*
>
> > Q3: Ablations on different augmentation ratios and component contributions.
>
> **A3:**
> We conducted ablations in Appendix E.3 to study the effects of varying the probabilities of substituting with query pairs ($p_q$) and random pairs ($p_r$), and to clarify the contributions of data augmentation versus the fusion architecture.
>
> 1. **Augmentation Ratios**: Balanced $p_q$ and $p_r$ are crucial. Excessive $p_q$ makes fusion task trivial and weakens robustness. Excessive $p_r$ introduces too much noise and prevents stable learning. Very small ratios offer little regularization. Our ablations show that intermediate values yield the best balance, with suitable ranges of $p_r \in [0.1, 0.3]$ and $p_q \in [0.1, 0.4]$.
>
> 2. **Locality-Aware Fusion vs. Augmentation**: Locality-aware fusion *play a more important role* than data augmentation. *Locality-aware fusion addresses comprehensive information extraction at the architectural level. Data augmentation primarily enhances stability and generalization.* Removing augmentation drops mIoU by 1.03%, while replacing locality-aware fusion with patch-wise fusion drops mIoU by 1.53%.
>
> These results show that data augmentation supports robustness. But the locality-aware design is most essential for extracting comprehensive, noise-resistant information from multiple prompts.

---

> > ### Author Response · Authors · 2025-11-25
> > **Author Response (2/2)**
> >
> > > Q4: Need to Discuss PromptHub's Limitations
> >
> > **A4:** We thank the reviewer for highlighting this. We have added a discussion in Appendix B.1 to explicitly address the main limitation of PromptHub: the **White-Box Dependency**. Like CONDENSER, PromptHub requires access to the backbone's parameters and gradients to train the fusion module, even though the backbone itself is frozen. This makes scaling to very large models or closed-source models challenging, as full gradients may be inaccessible or too costly. While this enables superior performance, extending prompt fusion to black-box or gradient-free settings is a key direction for future work.
> >
> > *We have added a discussion of this issue in Appendix of the revised manuscript.*
> >
> > > Q5: Analysis of PromptHub's Failure Cases
> >
> > **A5:** PromptHub struggles when retrieved prompts are spatially or semantically misaligned with the query. For example, the target object may be shifted far from its usual position or the background context may contradict the query. Locality-aware fusion assumes relevant context lies near corresponding query patches. Objects outside this receptive field may be treated as noise, causing the model to miss critical cues.
> >
> > This limitation points to two potential improvements. First, retrievers that consider both content and spatial layout can improve alignment, as pixel-level retrievers demonstrate (e.g. Prompt-SelF). Second, adaptive global attention could relax spatial constraints when local matches fail while maintaining enough regularization to prevent noise issues.

---

> ### Author Response · Authors · 2025-11-27
> **A Gentle Reminder of the Post-Rebuttal Feedback**
>
> Dear Reviewer `3dRG`，
>
> Thank you again for your time and valuable feedback on our paper. While we understand that you may have a busy schedule，we would like to confirm whether our response and revised paper have addressed your concerns.
>
> If you have any further questions or feel that we have misunderstood your feedback, please let us know. We are happy to discuss with you. Furthermore, if our responses and the revised paper have addressed your concerns, we would be grateful if you would consider updating your evaluation.
>
> Sincerely,
>
> The Authors

---

### Official Review · Reviewer_92Kf · 2025-11-01

**Soundness:** 3
**Presentation:** 3
**Contribution:** 2
**Rating:** 4
**Confidence:** 4

**Summary:**

Focues on the visual in-context learning, this paper proposes PromptHub, a novel learning-based framework to generate the optimal fused examples for each query from N examples. To do this, PromptHub first designs a locality-enhanced cross-attention strategy between the query image and examples to generate the unified prompt, and then develop three alignment loss to train the fusing process. Extensitve results on three vision tasks (segmentation, detection, and coorization) show the effectiveness of the proposed PromptHub.

**Strengths:**

1) The task of generating optimal prompt in VICL is a hot topic in the community. And the results of PormptHub show the improvements of this method.

2) Three new learning objectives are developed to train the fusing process.

**Weaknesses:**

1), One of the main concerns lies in the novelty of the work. Given that the previous CONDENSER model also generates fusing prompts from the query image and N pairs, I find that the locality-aware attention and training losses developed in PromptHub may limit the novelty of this paper. I hope the authors can clarify the core differences between the previous work and PromptHub in terms of the main motivation, fusing strategy, and training process.

2), Table 3 presents ablation results demonstrating the effectiveness of the designed modules, which helps readers understand their contributions. However, I notice that when using the global fusing strategy (a traditional approach), the performance of PromptHub is lower than that of CONDENSER. Does this suggest that the three newly introduced losses do not effectively improve alignment? The authors should provide a deeper discussion of this issue.

3), What is the motivation behind the three proposed losses, particularly L_s and L_u ? These losses might negatively affect the fusing process if their hyperparameters are not properly tuned.

4), Table 2 reports cross-dataset results of PromptHub, showing the strong transferability of the proposed model. I am also interested in its cross-task performance—specifically, when the model is trained on segmentation but tested on detection. This setting would be highly valuable to the VICL community, as it evaluates the model’s ability to perform in-context learning on unseen tasks.

**Questions:**

Please see the Weaknesses section above.

---

> ### Author Response · Authors · 2025-11-25
> **Author Response (1/3)**
>
> Dear reviewer `92Kf`,
>
> Thanks for the insightful comments and suggestions! We appreciate your positive reviews on the **good performance**. Below are our point-to-point clarifications to address your concerns. Feel free to discuss with us if you have further questions!
>
> > Q1: Novelty of PromptHub and its core differences from Condenser in terms of motivation, fusion strategy, and training process.
>
> **A1:**  Both PromptHub and Condenser rely on prompt fusion to leverage multiple prompts in VICL. However, PromptHub adopts a more comprehensive and robust fusion strategy, enhances the semantic quality of the fused prompt, and strengthens the backbone’s reliance on it, aspects that Condenser does not address. PromptHub therefore delivers more reliable and effective prompt fusion, and this advantage is consistently supported by cross-dataset and cross-task evaluations, visual analyses, and quantitative measurements.
>
> The core differences are threefold:
>
> 1. **Motivation: Prompt Fusion vs. (More Reliability and Effectiveness) Prompt Fusion**
>
> **Condenser (Previous Work):** It aimed to compress multiple prompts to avoid the computational cost of ensembles or the information loss of downscaling. However, it ignored the quality of the fused representation. Figure 6 shows Condenser produces noisy, semantically incoherent patterns, which may lead the backbone to ignore the fused prompt and rely on its internal priors.
>
> **PromptHub (Ours):** Our motivation is to achieve more reliable and effective prompt fusion. We found that rigid fusion causes feature misalignment. Our goal is to create a fused prompt that is concise, semantically reliable, and query-aligned, encouraging the backbone to utilize in-context cues.
>
> 2. **Fusion Strategy: Rigid Patch-wise vs. Locality-Aware**
>
> **Condenser:** Uses patch-wise cross-attention, aggregating information only at identical spatial coordinates $(h,w)$. It assumes perfect alignment. Any shifted semantic feature is missed, giving a zero receptive field beyond one patch.
>
> **PromptHub**: Introduces locality-enhanced fusion via a spatial prior (Gaussian/Laplacian). By optimizing $\sigma$, it generalizes Condenser ($\sigma \to 0$) to create a soft receptive field that captures neighboring context, reduces noise, and improves visual coherence (see Figure 6).
>
> 3. **Training Objectives: Input Regularization vs. "Fusion-Utilization-Prediction" Chain**
>
> **Condenser:** Relies on a simple Pre-Alignment loss ($\mathcal{L}_{PA}$) before the backbone. This only aligns input features but does not guarantee the correctness of their deep semantic information.
>
> **PromptHub:** Implements a comprehensive "fusion-utilization-prediction" chain. We introduce two novel objectives specifically designed to enhance fused prompt quality and ensure backbone "trust" on it. To assist the proposed loss, we also design data augmentation strategies that randomly swap prompts with query or random pairs during training to purify utilization signals and enhance PromptHub robustness.
>
> **Summary:** PromptHub and Condenser both focus on prompt fusion, but PromptHub further addresses key limitations that remain unresolved in Condenser.
>
> 1. PromptHub employs locality-enhanced fusion, enabling broader context aggregation and more effective noise suppression than Condenser.
>
> 2. PromptHub introduces a “fusion–utilization–prediction” training scheme that improves fused-prompt quality and increases backbone reliance on it, aspects not handled by Condenser.
>
> 3. Extensive experiments demonstrate the significance of PromptHub, showing stronger performance across cross-task and cross-dataset evaluations and delivering more reliable prompt fusion for VICL.
>
> We have revised the abstract, introduction, and experimental sections to strengthen these points.

---

> > ### Author Response · Authors · 2025-11-25
> > **Author Response (2/3)**
> >
> > > Q2: Why PromptHub underperforms Condenser when using global fusion strategy.
> >
> > **A2:**
> > We appreciate this detailed observation. We respectfully clarify that the lower performance of our "Global Fusion" variant compared to the "Full CA" variant in Condenser does not stem from the ineffectiveness of our proposed losses ($\mathcal{L}_p$, $\mathcal{L}_s$, $\mathcal{L}_u$). Instead, it is primarily due to the inherent structural disadvantage of global fusion in this context, combined with *differences in experimental settings*:
> >
> >   1. *Training Epoch Discrepancy*: All PromptHub variants (including "Global Fusion") were trained for **100 epochs**, whereas the Condenser paper reports **150 epochs**. Because global attention has a larger search space, it generally requires more iterations to converge. Despite this 33% reduction in training time, our "Global Fusion" (N=16) achieves 44.64% mIoU , which is comparable to Condenser's "Full CA" ablation result of 44.87%.
> >
> >   2. *Implementation Differences*: We implemented "Global Fusion" by setting our locality parameter $\sigma$ to an extremely large value to simulate a flat distribution. This differs from Condenser specific "Full CA" implementation.
> >
> > We further conducted "Global Fusion" performance on PromptHub under the same settings as Condenser, and the resulting performance is reported below.
> > |    Method            |     Fold-0    | Fold-1 | Fold-2 | Fold-3 |   Avg |Det. |
> > |-|-|-|-|-|-|-|
> > |  Condenser (Full CA) |     44.00     |  49.97 | 42.55  | 42.97  |  44.87|43.11|
> > |  PromptHub (Full CA) |     44.74     |  51.66 | 44.91  | 44.25  |  46.39|43.83|
> >
> > Under the fair comparison, the "Full CA" ablation of PromptHub still outperforms Condenser, indicating that our three proposed losses remain effective even in the global fusion setting.
> >
> > > Q3: Motivation behind the three proposed losses and potential negative influence on fusion if not properly tuned.
> >
> > **A3:** The design of our three losses is not arbitrary, they form a cohesive "fusion-utilization-prediction" chain necessary to address specific failure modes we observed in multi-prompt VICL.
> >
> > The specific motivations are as follows:
> >
> > 1. $\mathcal{L}_p$ (Label Prediction Loss): The label prediction loss serves as a fundamental objective to preserve the contextual prediction capability in VICL. It is also a standard component in prior methods such as Condenser and InMeMo; without this loss, a training-based VICL framework with additional learnable parameters cannot function properly. This loss is a standard component rather than part of our contributions.
> >
> > 2. $\mathcal{L}_s$ (Semantic Integrity Loss): Ensuring Fused Prompt Quality
> >
> >   - Motivation: We observe that Condenser's input-level alignment may only enforce a superficial correspondence between fused prompt features and query features, while deeper semantic representations remain insufficiently aligned. We operate on the premise that the ideal prompt for a query is one that is semantically identical to the query itself.
> >
> >   - Mechanism: Therefore, $\mathcal{L}_s$ aligns the continuous tokens of the fused prompt with the discrete tokens of the query pair. This ensures the fused prompt is semantically meaningful and aligned with the query's content, promoting high-quality fusion.
> >
> > 3. $\mathcal{L}_u$ (Utilization Loss): Ensuring Backbone Reliance on Prompt
> >
> >   - Motivation: Due to the discrepancy between the fused prompt and a natural query image, the pre-trained backbone may treat the information-enriched fused prompt as unreliable noise and fall back on its internal priors (ignoring the context cues).
> >
> >   - Mechanism: $\mathcal{L}_u$ explicitly minimizes the feature discrepancy (via cosine similarity) between the fused prompt and the query pair. This forces the fused representation to reside in the same feature manifold as the query, compelling the backbone to effectively trust and utilize the prompt cues during inference.
> >
> > We acknowledge that multi-objective optimization requires balancing, as excessive emphasis on $\mathcal{L}_s$ or $\mathcal{L}_u$ may degrade other objectives. Our ablation study (Table 3) confirms that the three losses are complementary: $\mathcal{L}_s$ ensures the prompt quality, $\mathcal{L}_u$ ensures backbone utilization, and $\mathcal{L}_p$ ensures correct predictions. A sensitivity analysis in Appendix E.2 (Figure 10) shows that a desirable balance is achieved when $\lambda \in [0.2, 0.6]$ and $\gamma \in [0.1, 0.3]$.
> >
> > The above explanations are also discussed in the main paper (lines 51–88 and lines 250–265).

---

> > > ### Author Response · Authors · 2025-11-25
> > > **Author Response (3/3)**
> > >
> > > > Q4: Investigation of cross-task performance for PromptHub versus Condenser.
> > >
> > > **A4:**
> > > We thank you for proposing this valuable evaluation setting. We agree that cross-task transfer is a rigorous test of a model's intrinsic VICL capabilities.
> > >
> > > To address this, we conducted an experiment where models trained solely on segmentation (Pascal-5i, four folds) were directly tested on detection (Pascal VOC 2012) without any fine-tuning. We compared PromptHub against the CONDENSER baseline (using their open-source checkpoints). The results are reported below:
> > > |    Method      |     Fold-0    | Fold-1 | Fold-2 | Fold-3 |   Avg |
> > > |-|-|-|-|-|-|
> > > |  Condenser$_{N=1}$ |     38.15     |  35.70 |  35.49 |  30.31 | 34.91 |
> > > | Condenser$_{N=16}$ |     41.25     |  36.66 |  37.86 |  39.02 | 38.70 |
> > > |  PromptHub$_{N=1}$ |     41.59     |  36.25 |  37.71 |  32.15 | 36.93 |
> > > | PromptHub$_{N=16}$ |     43.40     |  38.55 |  39.52 |  40.66 | 40.53 |
> > >
> > > PromptHub consistently outperforms Condenser in this challenging cross-task setting. Notably, with N=16, PromptHub achieves a +1.83% mIoU improvement over the baseline. This indicates that our locality-aware fusion strategy captures more robust, fundamental visual cues that generalize better to unseen tasks than the patch-wise approach.
> > >
> > > *We note that although the overall performance is strong, the training process is not fully task-agnostic. Because the model is trained to reconstruct segmentation masks, a domain gap naturally arises when transferring to bounding box detection.*
> > >
> > > We have added a discussion of this issue in Section 4 of the revised manuscript.

---

> ### Author Response · Authors · 2025-11-27
> **A Gentle Reminder of the Post-Rebuttal Feedback**
>
> Dear reviewer `92Kf`，
>
> Thank you again for your time and valuable feedback on our paper. While we understand that you may have a busy schedule，we would like to confirm whether our response and revised paper have addressed your concerns.
>
> If you have any further questions or feel that we have misunderstood your feedback, please let us know. We are happy to discuss with you. Furthermore, if our responses and the revised paper have addressed your concerns, we would be grateful if you would consider updating your evaluation.
>
> Sincerely,
>
> The Authors

---

### Official Review · Reviewer_yUCw · 2025-11-03

**Soundness:** 3
**Presentation:** 3
**Contribution:** 3
**Rating:** 6
**Confidence:** 3

**Summary:**

This paper focuses on enhancing the multi-prompting capability of MAE-VQGAN through the method of prompt fusion. The authors identify two main limitations of the recent prompt-fusion method CONDENSER: it fails to fully utilize local visual cues from multiple prompts, and the discrepancies that arise from prompt fusion can cause the model to revert to its standard inference method. To address these issues, the authors introduce PromptHub, which spatially aggregates relevant information from multiple prompt pairs using cross-attention. Additionally, they implement learning objectives and data augmentation for prompt pairs to help mitigate the discrepancies. Experiments conducted on different benchmarks demonstrate the effectiveness of the proposed method.

**Strengths:**

- The concept of utilizing local-aware aggregation for prompt fusion and the design of learning objectives is illustrative and aligned with the original motivation.

- The performance improvement is significant compared to competitors.

- The writing is generally well-structured and easy to understand.

**Weaknesses:**

- **Interpretability**: As shown in Figure 6, the fused prompts of Condenser align closely with the target regions of the queries. In contrast, the generated object contours in PromptHub are misaligned with the queries. To improve clarity, it would be beneficial to include additional visual results, such as attention maps of the prompt regions similar to those in Figure 13.

- The cross-attention operation is constrained by the coordinate position. What would happen if the prompt pairs have distinct positions compared to the query? I wonder if this situation would harm the model's performance.

**Questions:**

Please address the concerns mentioned in the weaknesses section.

In Section 3.4 (ii), random pairs are sampled as prompt pairs to improve the robustness of PromptHub. I am curious if the authors have assessed the model's performance when these prompt pairs are combined with noisy pairs.

---

> ### Author Response · Authors · 2025-11-25
> **Author Response (1/2)**
>
> Dear reviewer `yUCw`,
>
> Thank you for the thoughtful feedback and constructive suggestions! We appreciate your positive reviews on **clear motivation** and **effective designs**. Below are our detailed responses to your remaining questions.
>
> > Q1: Interpretability and prompt-fusion alignment compared to Condenser baseline
>
> **A1:**
> According to Figure 6，Condenser appears to align with target regions. Our analysis suggests its "fused prompt" images primarily capture low-level features like lighting or contrast rather than semantic content. For example, in the 5th image from the left in Figure 6, Condenser merely highlights the brightest area (the white box) rather than the semantic query object. Furthermore, Condenser's "fused prompt" labels often manifest as ambiguous halos centered in the image, lacking specific semantic shape information.
>
> To strictly measure the alignment between the "fused prompt" and the actual query, we calculated the IoU between the "fused prompt label" and the "query label (ground truth)". This metric objectively evaluates how well the fused knowledge aligns with the target object.
>
> As shown in the table below, PromptHub significantly outperforms Condenser:
> | Method             | Fold-0 | Fold-1 | Fold-2 | Fold-3 | Mean    |
> |--------------------|--------|--------|--------|--------|--------|
> | Condenser$_{N=1}$  | 18.93  | 29.73  | 24.26  | 27.94  | 25.22 |
> | Condenser$_{N=16}$ | 14.27  | 24.56  | 18.85  | 20.56  | 19.56  |
> | PromptHub$_{N=1}$  | 21.25  | 37.61  | 35.01  | 29.79  | 30.92 |
> | PromptHub$_{N=16}$ | 25.22  | 43.22  | 36.07  | 30.41  | 33.73  |
>
> *We have added this experiment to Section 4 of the revised manuscript.*
>
> PromptHub(N=16) achieves an average IoU of 33.73, an absolute gain of 14.17 points and a relative improvement of approximately 72% over Condenser(N=16) (19.56).
>
> This quantitatively demonstrates that PromptHub generates fused prompts that are semantically closer to the query targets.
>
> *Additional Visualizations (Appendix Update)*: Following your suggestion to improve clarity, we have added more visual comparisons, including attention maps of the prompt regions similar to Figure 13, to Appendix Section I.2. We can observe that most of the relevant patches are correctly attended to.
>
> *Interpretable*: For clarity, we have revised the terminology in the manuscript by replacing *interpretable* with *reliable* to better reflect the intended meaning.
>
> > Q2: Position shift and its potential impact on PromptHub's performance.
>
> **A2:**
> **We appreciate the reviewer raising the important question regarding the impact of spatial misalignment between prompts and queries.**
>
> We acknowledge that spatial misalignment between prompts and queries may negatively affect prompt fusion performance. However, we believe that PromptHub's locality-aware fusion can better alleviate such spatial misalignment compared with the patch-wise fusion used in Condenser.
>
> To evaluate performance under position shifts, we conducted an experiment where query pairs were horizontally flipped and retrained to simulate severe spatial misalignment between the retrieved prompts and the query image. We compared the performance drop of Condenser and PromptHub under the perturbed conditions.
> |Method|N|Standard mIoU (Mean)| Perturbed mIoU (Mean) | Performance Drop |
> |-----|---|-------------------|-----------------|------------------|
> | Condenser | $N=1$  | 44.14| （40.35+48.32+40.88+39.87）/4 =42.36| -1.78|
> | Condenser | $N=16$ | 46.63| （43.80+51.07+42.59+43.48）/4 =45.24| -1.39|
> | PromptHub | $N=1$  | 45.71| （42.66+49.84+42.49+42.13）/4 =44.28| -1.43|
> | PromptHub | $N=16$ | 48.10| （45.96+52.25+45.21+45.43）/4 =47.21| -0.89|
>
> The experiment yields the following observations:
>
> 1. PromptHub experiences a smaller performance drop than Condenser, mainly due to its locality-wise fusion mechanism. The patch-wise fusion of Condenser enforces strict spatial correspondence and is therefore highly sensitive to positional shifts, while locality fusion allows PromptHub to aggregate relevant semantics even under spatial misalignment.
>
> 2. Furthermore, increasing the number of prompts ($N$) further alleviates misalignment, since a larger prompt pool increases the likelihood of prompt pairs with more similar poses or positions to the query.
>
> *We have added a discussion of this issue in Section 4 of the revised manuscript.*

---

> > ### Author Response · Authors · 2025-11-25
> > **Author Response (2/2)**
> >
> > > Q3: Effect of introducing random noisy prompt pairs during inference on VICL performance.
> >
> > **A3:** We conducted a controlled study to evaluate whether injecting random noisy prompt pairs at inference degrades model performance. Specifically, we compared results across different noise injection ratios and against a setting without the augmentation described in Section 3.4(ii). The results show:
> > |Method|noise ratio|Fold-0 | Fold-1 | Fold-2 | Fold-3 |Mean|
> > |-----|---|--------|--------|--------|--------|--------|
> > | PromptHub$_{N=16}$|0%|46.68|53.08|46.15|46.52|48.10|
> > | PromptHub$_{N=16}$|10%|46.34|52.82|46.21|46.67|48.01|
> > | PromptHub$_{N=16}$|25%|46.23|52.90|45.94|46.25|47.83|
> > | PromptHub$_{N=16}$|50%|44.12|51.39|44.98|44.17|46.17|
> > | PromptHub$_{N=16}$|100%|42.26|49.67|42.92|41.45|44.08|
> > | w/o augmentation$_{N=16}$|0%|45.84|52.01|44.83|45.60|47.07|
> > | w/o augmentation$_{N=16}$|10%|45.16|51.43|44.24|44.03|46.22|
> > | w/o augmentation$_{N=16}$|25%|44.28|51.19|43.85|43.16|45.62|
> > | w/o augmentation$_{N=16}$|50%|43.27|50.57|43.00|41.86|44.68|
> > | w/o augmentation$_{N=16}$|100%|42.09|49.45|42.69|39.73|43.49|
> >
> > 1. We observe that the augmentation strategy plays a critical role in maintaining robustness: with 10% and 25% noise ratios, the model remains relatively strong, while performance starts to degrade when noise reaches 50% or even 100%.
> >
> > 2. In contrast, when removing the augmentation strategy during training, performance drops even at 10% and 25% noise ratios, and remains consistently lower than PromptHub when noise ratios reach 50% and 100%.
> >
> > 3. These results clearly validate the importance of our data augmentation design.
> >
> > *We have added a discussion of this issue in Section 4 of the revised manuscript.*

---

> ### Author Response · Authors · 2025-11-27
> **A Gentle Reminder of the Post-Rebuttal Feedback**
>
> Dear reviewer `yUCw`，
>
> Thank you again for your time and valuable feedback on our paper. While we understand that you may have a busy schedule，we would like to confirm whether our response and revised paper have addressed your concerns.
>
> If you have any further questions or feel that we have misunderstood your feedback, please let us know. We are happy to discuss with you. Furthermore, if our responses and the revised paper have addressed your concerns, we would be grateful if you would consider updating your evaluation.
>
> Sincerely,
>
> The Authors

---

### Author Response · Authors · 2025-11-25
**General Responses and Revision Log**

Dear Reviewers and Chairs,

We sincerely appreciate the reviewers' constructive feedback, which has been highly valuable for improving our manuscript. We also thank the Chairs for their time of our submission.

We are encouraged by the recognition that our paper is **well-written with clear motivation** (`yUCw`, `3dRG`), our task is a **hot topic** (`92Kf`), and the performance improvement is **significant** (`yUCw`, `92Kf`, `3dRG`).

Summary of our contributions:
- We propose locality-enhanced fusion, which generalizes rigid patch-wise matching to a locality-aware mechanism, enabling robust context aggregation and effective noise suppression.
- We introduce a "fusion-utilization-prediction" training scheme that enhances the fused prompt quality and ensures the backbone actively leverages the information-enriched fused prompts.
- Extensive experiments demonstrate PromptHub's superiority, particularly in challenging scenarios such as cross-dataset transfer, and spatially misaligned settings.
- PromptHub advances prompt fusion toward a more reliable and effective paradigm, providing VICL with more robust multi-prompt integration and stronger cross-task generalization capabilities.

We have updated the manuscript in response to the concerns raised during the rebuttal, and the key revisions are summarized below.
- We have added citations and discussions of the latest related works, including some concurrent VICL submissions to ICLR '26.
- To avoid ambiguity, we replaced every instance of “interpretable” with “reliable.”
- We added cross-task transferability experiments comparing Condenser and PromptHub in Section 4.3 (vi).
- We added experiments that evaluate both Condenser and PromptHub under spatial misalignment in Section 4.3 (vii).
- We included results on PromptHub’s behavior under varying levels of random-pair injection in Section 4.3 (viii).
- We added a quantitative analysis in Section 4.4 that measures the similarity between fused prompt labels and query labels to characterize “reliable” fused prompts.
- We documented the limitations of our approach in Appendix B.1.
- We included a hyperparameter study for $p_r$ and $p_q$ in Appendix E.3.
- We provided additional fusion attention visualizations for PromptHub in Appendix I.2.

We sincerely thank all reviewers again for their valuable feedback. If there are any remaining or new concerns, we will do our best to address them in detail.

Best,

Authors

---

### Author Response · Authors · 2025-11-28
**Welcome to comment during the OpenReview API security incident!**

Dear Reviewers and Chairs,

We were sorry to learn that OpenReview has just suspended review editing privileges due to the recent API security incident. We understand that, as a result, you may currently be unable to update your official reviews.

Fortunately, we find the comment function remains active. We would be very happy to clarify any remaining questions or discuss any points that remain unclear. We would greatly appreciate the reviewers' input and evaluation of our added results and improvements to the paper.

We also wish you the best of luck with your own submissions and hope everything proceeds smoothly for you.

Sincerely,

The Authors

---

### Author Response · Authors · 2025-12-02
**Summary of Rebuttal Phase**

Dear Area Chair,

We are grateful for your time and dedication in handling our submission. We have carefully revised the manuscript to incorporate all the reviewers' constructive suggestions. To facilitate your final decision-making, we provide a brief summary below detailing how our updates effectively resolve the primary concerns and further solidify the contributions of PromptHub.

**Summary of Our Contributions:**

- We propose a locality-enhanced fusion that extends rigid patch-wise to a locality-aware formulation, enabling more robust context aggregation and improved robustness.
- We introduce a "fusion-utilization-prediction" training scheme that improves fused prompt quality and encourages the backbone to effectively exploit it.
- Extensive experiments demonstrate superiority, particularly in challenging settings such as cross-dataset transfer, cross-task transfer, and spatial misalignment.
- Overall, PromptHub advances prompt fusion toward a more reliable and effective paradigm, equipping VICL with stronger multi-prompt integration and markedly improved fused prompt accuracy.

**Resolve the Primary Concerns:**

| Reviewer | Strengths | Key concerns | Our response |
|:---|:---|:---|:---|
| **92Kf (4)** | • **Significance:** Important research topic | 1. PromptHub vs. Condenser | 1. Clarified distinctions in detail from 3 aspects: *motivation*, *mechanism*, and *training strategy* (**A1**). |
| | • **Soundness:** Three new objectives | 2. Performance diff with "Global Fusion" | 2. Listed setting differences (e.g., epoch number) and showed PromptHub outperforms (*46.39% > 44.87%*) Condenser under aligned settings. (**A2**) |
| | • **Effectiveness:** Stable performance gains | 3. Loss motivation | 3. Explained the *"fusion-utilization-prediction"* synergy of our three losses. (**A3**) |
| | | 4. Cross-task transfer | 4. Demonstrated **superior transferability** (Seg $\rightarrow$ Det): outperforms Condenser (*40.53% > 38.70%* in **Tab. 5**) (**A4**). |
| **yUCw (6)** | • **Soundness:** Clear motivation/design | 1. Fusion visualization | 1. Added alignment metrics (*+72% gain* in **Tab. 8**) & more attention maps **(Fig. 14)**. (**A1**) |
| | • **Effectiveness:** Consistent gains | 2. Performance under spatial mismatch | 2. Demonstrated *substantial robustness* over Condenser under spatial misalignment (*47.21% > 45.24%* in **Tab. 6)**. (**A2**)|
| | | 3. Noise injection ratios | 3. Verified *strong performance* retention under 10%-25% noise ratios (**A3**, also **Tab. 7**)|
| **3dRG (6)** | • **Soundness:** Well-motivated design | 1. Quantifying interpretability | 1. Quantified fused label accuracy (*+72% gain*, **Tab. 8**) and refined terminology to "*reliable*". (**A1**)|
| | • **Effectiveness:** Solid empirical results | 2. Fusion module design | 2. Clarified full-image fusion is a *deliberate design* for VICL versatility (e.g. image colorization, style transfer), not an oversight. (**A2**)|
| | | 3. Augmentation ablations, limitation discussion. | 3. Added augmentation studies **(Fig. 11)** & limitation's discussion **(Appendix.B.1)**. (**A3, A4, A5**)|

The updated manuscript now presents solid empirical evidence substantiating PromptHub's reliability, transferability, and robustness, as well as the effectiveness of its design. We trust that these efforts resolve all reviewers' concerns and improve the readability for publication.

Sincerely,

Authors

---

### Meta-Review · Area_Chair_UC2j · 2026-01-05

**Summary:**

This paper addresses the limitations of rigid patch-wise fusion in Visual In-Context Learning and proposes PromptHub, a framework featuring locality-aware fusion and a "fusion-utilization-prediction" training chain to improve alignment and robustness. Most reviewers found the proposed method effective but raised concerns regarding novelty against Condenser (Reviewer 92Kf), the interpretability of the fused prompts (Reviewers yUCw, 3dRG), and specific experimental performance comparisons (Reviewer 92Kf).

After the rebuttal, the AC finds that most of the questions and demands from the reviewers were addressed with extensive additional results provided during the process. The AC agrees that this work is an incremental follow-up to Condenser; however, weighing the effectiveness of the proposed locality-aware fusion and training paradigm against the incremental design, the AC recommends acceptance since the spatial-aware fusion could benefit the VICL community.

**Reviewer Concerns:**

Reviewer 92Kf has concerns about the novelty relative to Condenser and the performance validity. The authors clarified that the ablation discrepancy was due to training epoch settings and, crucially, addressed the generalization concern by adding the requested cross-task experiments. The AC finds that the superior performance effectively validates the robustness benefits of the proposed method over the baseline.

Reviewers yUCw and 3dRG both express worry about the interpretability and alignment quality of the fused prompts. The authors responded by refining the terminology to "reliable" and provided a new quantitative experiment to validate the improvement in semantic alignment compared to Condenser. The AC considers this evidence sufficient to resolve the concern regarding the fusion mechanism's effectiveness.

However, the AC finds some minor concerns remain. While the authors discussed failure cases in the rebuttal, the method inherently struggles when retrieved prompts are severely spatially misaligned, which remains a limitation of the locality-aware fusion design.

**Reviewer Scores:**

Reviewer yUCw: 6. Reviewer 92Kf: 4. Reviewer 3dRG: 6.

The AC believes that Reviewers yUCw and 3dRG would highly likely maintain their scores given the rebuttal. Furthermore, since most of the concerns raised by Reviewer 92Kf have been addressed, the AC considers that the evaluation merits a score increase (or at least a stronger inclination towards acceptance).

---

### Decision · Program_Chairs · 2026-01-26

Accept (Poster)